# High antibody levels and reduced cellular response in children up to one year after SARS-CoV-2 infection

Eva-Maria Jacobsen[1,15], Dorit Fabricius [1,15], Magdalena Class [1], Fernando Topfstedt[2], Raquel Lorenzetti[2], Iga Janowska[2], Franziska Schmidt[2], Julian Staniek[2], Maria Zernickel [1], Thomas Stamminger [3], Andrea N. Dietz[3], Angela Zellmer[4], Manuel Hecht[4], Peter Rauch[4], Carmen Blum[1], Carolin Ludwig[5,6], Bernd Jahrsdörfer [5,6], Hubert Schrezenmeier[5,6], Maximilian Heeg [7], Benjamin Mayer [8], Alina Seidel[9], Rüdiger Groß [9], Jan Münch [9], Frank Kirchhoff [9], Sebastian F. N. Bode [1], Gudrun Strauss[1], Hanna Renk [10], Roland Elling [11,7], Maximillian Stich [12], Reinhard E. Voll[2,7], Burkhard Tönshof[12], Axel R. Franz [10], Philipp Henneke [11,7], Klaus-Michael Debatin[1], Marta Rizzi [2,13,14] ✉ & Ales Janda [1] ✉

The COVID-19 course and immunity differ in children and adults. We analyzed immune response dynamics in 28 families up to 12 months after mild or asymptomatic infection. Unlike adults, the initial response is plasmablast-driven in children. Four months after infection, children show an enhanced specific antibody response and lower but detectable spike 1 protein (S1)-specific B and T cell responses than their parents. While specific antibodies decline, neutralizing antibody activity and breadth increase in both groups. The frequencies of S1-specific B and T cell responses remain stable. However, in children, one year after infection, an increase in the S1-specific IgA class switch and the expression of CD27 on S1-specific B cells and T cell maturation are observed. These results, together with the enhanced neutralizing potential and breadth of the specific antibodies, suggest a progressive maturation of the S1-specific immune response. Hence, the immune response in children persists over 12 months but dynamically changes in quality, with progressive neutralizing, breadth, and memory maturation. This implies a benefit for booster vaccination in children to consolidate memory formation.

Infection with severe acute respiratory syndrome coronavirus-2 (SARS-CoV-2) has a mild or asymptomatic course in most individuals, particularly children and young adults[1,2]. SARS-CoV-2 viral components, namely, the spike (S) protein, the nucleocapsid protein (NCP), the membrane (M) protein and viroporins (ORF3a and ORF7a), are highly immunogenic and induce robust T cell, B cell and antibody responses.

However, the persistence of SARS-CoV-2 immunity following mild/asymptomatic infections in adults is still under investigation, and information on children is even less complete[3–7]. In adults, six-eight months post infection, antibody titers to S1 and NCP progressively decline[8]. At the same time, the quality of the antibodies improves with an increase in neutralizing activity and in their ability to recognize

variants of concern (VOC), indicating a progressive maturation in terms of breadth and affinity of the antibody response, which also continues when the infection is resolved[9–12]. Circulating memory B cells specific for SARS-CoV-2 emerge in adults two to three weeks after infection and persist for up to six months. In contrast, CD4[+] and CD8[+] T cell responses decline over time[8].

The understanding of the durability of SARS-CoV-2-specific adaptive immune responses in children is relevant, given their unique, i.e. more frequently asymptomatic disease course and their particular position in vaccination campaigns. In this work, families with at least one adult or child with confirmed mild/asymptomatic SARS-CoV-2 infection were followed for up to 12 months after the infection and analyzed for the persistence of SARS-CoV-2-specific antibodies (anti-S, -RBD and -NCP) and specific B and T cells (S1 and RBD). As controls, we used family members without a history of SARS-CoV-2 infection. Our data show that in children, the initial response is driven by plasmablasts and antibodies with a low frequency of virus-specific memory B and T cells. SARS-CoV-2-specific immunity persists for over 12 months both in children and adults and shows signs of phenotypic maturation in both the B and T cell compartments.

## Results
### Cohort
Within the previously described southwest household cohort[13,14], we studied 28 families, with 61 adults (median age 44.8 years, IQR 41.4–50.0) and 50 children (median age 10.4 years, IQR 7.2–13.5). We selected families with at least one child or adolescent and one COVID-19 case either assessed by polymerase chain reaction (PCR) or by seroconversion (see Methods for inclusion criteria and definition of seropositivity). Among those, 31 adults (50.8%) and 27 children (54.0%) were seropositive at the time of study inclusion. In 93.5% of infected adults and in 66.7% of infected children, the disease course was mild, and in 6.5% of adults and 33.3% of children, infection was classified as asymptomatic. None of the participants were hospitalized. As shown in Table 1, the most frequent symptoms were fever, cough and dysgeusia. Sampling was performed during national contact restriction (lockdown) in the summer of 2020 and afterwards in spring 2021, corresponding to approximately 4 and 12 months after infection, respectively.

### Decrease in antibody titer but progressive neutralizing potency after SARS-CoV-2 mild/asymptomatic infection in children and adults
We studied the specific antibody responses (IgG, IgA) to the S1 subunit of the spike protein (Fig. 1A, B), the receptor-binding domain (RBD) (Supplementary Fig. 1A) and NCP (Supplementary Fig. 1B). In line with previous analyses of this cohort[13,14] and of other cohorts[4], 4 months after infection, serum titers of specific S1 antibodies were higher in children than in adults (Fig. 1A). The difference between the two groups disappeared 12 months after infection (Fig. 1A). In contrast, the specific IgG antibody response to RBD protein was similar between adults and children over the observation period (Supplementary Fig. 1A). Notably, a significantly higher response to NCP was observed in children 4 months after infection but rapidly decreased, becoming significantly lower in children compared to adults 12 months after infection (Supplementary Fig. 1B). For IgA, S1-specific responses were similar between children and adults in the observation period (Fig. 1B). Overall, we observed a progressive decrease in the S1 (Fig. 1A), RBD (Supplementary Fig. 1A) and NCP-specific (Supplementary Fig. 1B) IgG antibodies in both adults and children. The extent of the decrease in S1 and RBD-specific antibodies between 4 months and 12 months after infection was similar in children and adults (Fig. 1A and Supplementary Fig. 1A), while children showed a much faster decrease in NCP-specific antibodies (Supplementary Fig. 1B). Interestingly, a progressive increase in S1-specific IgA antibodies was observed in children but not in the adult population (Fig. 1B). These data point to progressive class switching over time in B cells of the pediatric cohort. With respect to the functionality of the antibodies, we found neutralizing antibodies in both cohorts; however, children showed a higher neutralization capacity than adults (Fig. 1C). Neutralizing antibodies were still detectable 12 months post infection in both cohorts, even though they significantly decreased in the adult population (Fig. 1C). Neutralizing antibodies mostly bind to RBD, with variable activity, depending on the epitope[15]. Therefore, we used the neutralization potency index (NPI), defined as the relative proportion of plasma neutralization capacity with respect to the specific RBD antibody level[11,16]. The NPI (Fig. 1D) progressively increased in children and adults. Importantly, the differences in S1 and RBD-specific antibody levels between adults and

## Table 1 | Demographics and key information on the study participants

| Number of participants by age group (n) | Timepoint 1 (4 months) | | Timepoint 2 (12 months) | |
|---|---|---|---|---|
| | Adults (61) | Children (50) | Adults (51) | Children (40) |
| Median age (IQR) | 44.8 (41.4–50.0) | 10.4 (7.2–13.5) | 46.0 (41.6–51.1) | 10.7 (7.1–13.4) |
| Number of females (%) | 29 (47.5) | 26 (52.0) | 24 (47.1) | 19 (47.5) |
| BMI (IQR) | 25.4 (22.2–29.0) | 15.8 (14.5–19.4) | 26.3 (23.0–29.6) | 15.7 (14.5–19.4) |
| Number of seropositive participants (%) | 31 (50.8) | 27 (54.0) | 16/41 (39.0)* | 22 (55.0) |
| Asymptomatic (% of seropositive) | 2 (6.5) | 9 (33.3) | NA | NA |
| Persistent seropositivity at T2 (% of seropositive at T1) | NA | NA | 16/22 (77.3)*$ | 22/24 (91.7)** |
| Symptoms at disease onset (of seropositive participants) | | | | |
| Fever (%) | 18/31 (58.1) | 12/27 (44.4) | NA | NA |
| Cough (%) | 16/31 (51.6) | 6/27 (22.2) | NA | NA |
| Dysgeusia (%) | 16/31 (51.6) | 4/27 (14.8) | NA | NA |
| Diarrhea (%) | 7/31 (22.6) | 2/27 (7.4) | NA | NA |
| Vaccinated (%) | NA | NA | 10/51 (19.6)* | 0 |
| Participants with chronic disease: hypertension, diabetes mellitus, dyslipidemia (%) | 8/61 (13.1) | 0 | 7/51 (13.7) | 0 |

See methods for definition of how samples were defined as being seropositive, asymptomatic or symptomatic. Twenty-eight households with median of 114.0 days (IQR 109.3–120.0) since positive PCR test result (or symptom onset) and 23 households with median of 363.5 (IQR 361.8–372) were investigated at timepoint 1 (T1) and 2 (T2), respectively.
*BMI* body mass index, *IQR* interquartile range, *NA* not applicable, *PCR* polymerase chain reaction.
*Vaccinated donors were excluded from the analysis in T2.
**Three seropositive children and $five seropositive adults from T1 were lost to follow-up. Source data are provided as a Source Data file.

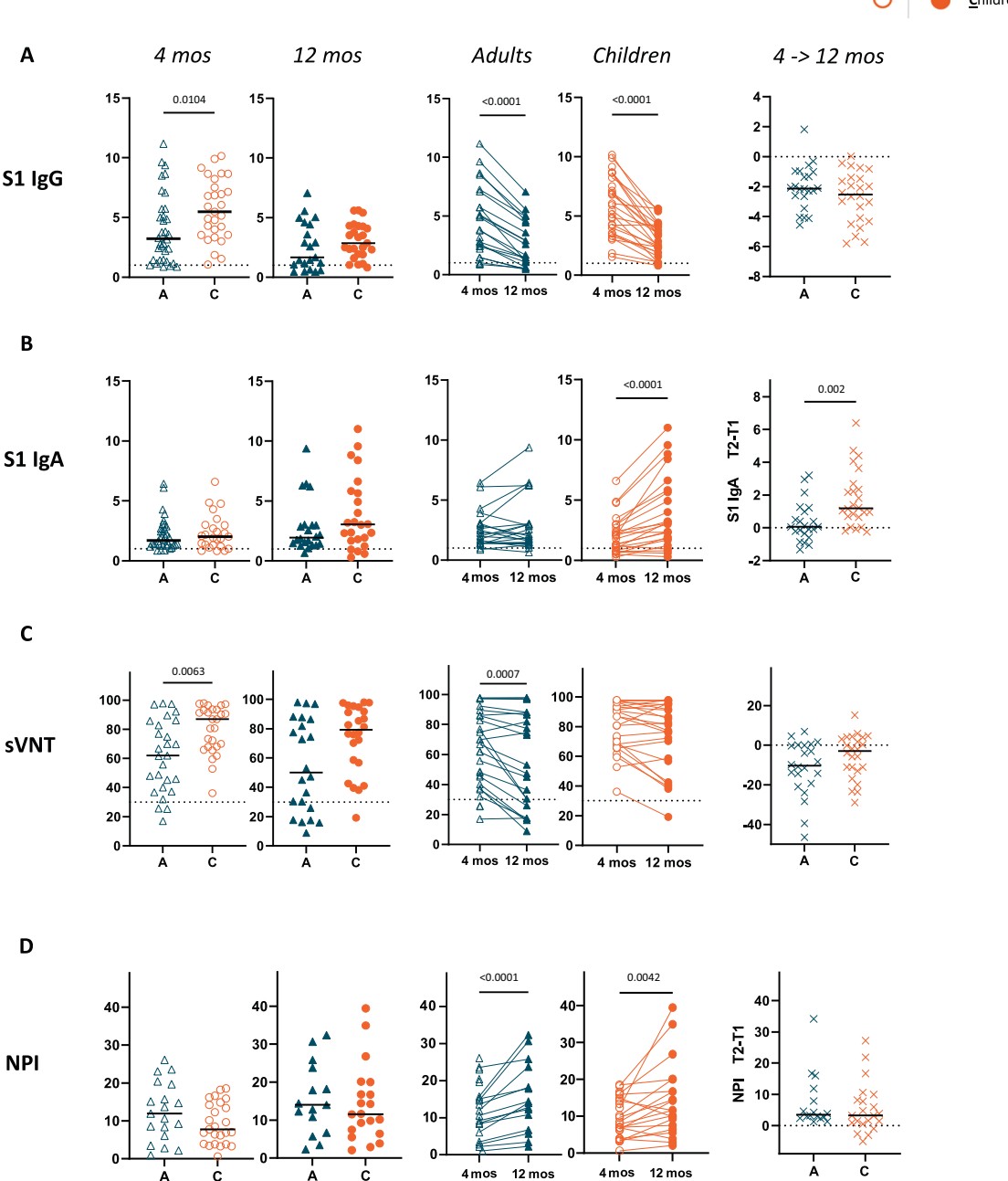

**Fig. 1 | Dynamics of the serum antibody response to SARS-CoV-2 in children and adults over 12 months.** Only participants who were seropositive at T1 are plotted (31 adults, 27 children; for definition of seropositivity see Methods). Antibodies reactive to SARS-CoV-2 spike 1 IgG (**A**) and IgA (**B**) were measured with (Euroimmun) ELISA at T1 (4 months after diagnosis; empty symbols) and T2 (12 months after diagnosis; filled symbols) in adults (A; blue symbols) and children (C; orange symbols); the dotted line represents a cut-off value for reactivity (cut-off index, COI = 1). **C** Neutralization serum capacity against the parental strain (WT, Wuhan) was tested with the GenScript SARS-CoV-2 Surrogate Virus Neutralization Test (sVNT). The cut-off value for neutralization (30%) is depicted with a dotted line. **D** The neutralization potency index (NPI) was calculated as sVNT/RBD IgG. In the far-right graphs, a difference between values at T2 and T1 is shown (ΔT2 − T1); the dotted line depicts a null difference. The Mann-Whitney test and Wilcoxon matched-pairs signed rank test were used for comparing median values (black lines) between adults and children and between T1 and T2, respectively. Only *p* values < 0.05 are shown. All the other p values, statistical analyses and source data are provided as a Source Data file.

children were not due to a change in total serum immunoglobulin levels (Supplementary Fig. 2A, B).

Sera of adults and children infected with the Wuhan wild-type form of the virus (WT) efficiently neutralized the variants of concern (VOC) alpha, beta and gamma 4 months after infection (Fig. 2A), even though neutralization potency gradually decreased from alpha to beta and gamma VOCs (Fig. 2A, B, Supplementary Fig. 3A). The neutralization capacity remained stable over the observation period

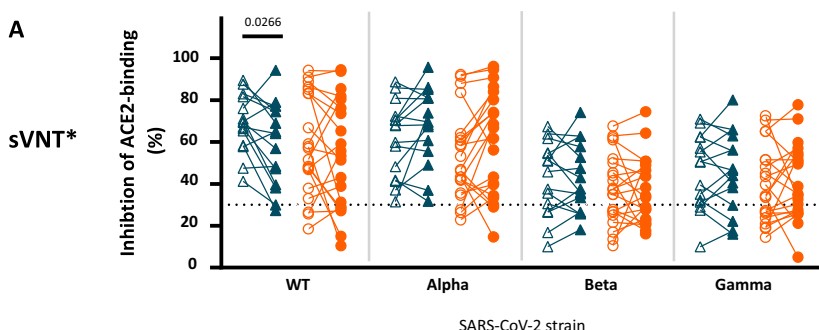

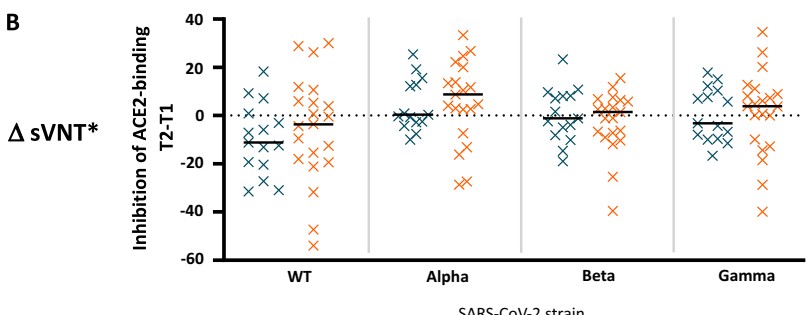

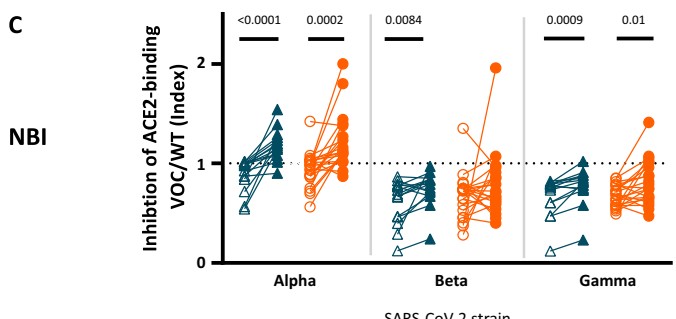

**Fig. 2 | Longitudinal decrease in neutralizing capacity against the original Wuhan (WT) strain with increasing neutralization breadth in children and adults.** Only participants who were seropositive at T1 (for definition of seropositivity see Methods) and who were tested in T2 (excluding vaccinated individuals) are plotted (16 adults, 21 children). **A** Neutralization serum capacity against parental (WT, Wuhan) and Variants of Concern (VOC): Alpha (B.1.1.7), Beta (B.1.351) and Gamma (P.1) strains was tested with the AcroNT SARS-CoV-2 Surrogate Virus Neutralization Test (sVNT*) at T1 (4 months after diagnosis; empty symbols) and T2 (12 months after diagnosis; filled symbols) in adults (**A**; blue symbols) and children (**C**; orange symbols); dotted line represents a cut-off value for neutralization (30%).

**B** Difference (Δ) in neutralization serum capacity between T2 and T1 (T2-T1) is plotted. **C** The neutralization breadth index (NBI) at T1 and T2 was calculated as the neutralizing activity of a particular VOC relative to the parental strain WT and plotted for T1 and T2 (the dotted line represents the same neutralization capacity against VOC as against WT). The Mann-Whitney test and Wilcoxon matched-pairs signed rank test were used for comparing median values (black lines) between adults and children and between T1 and T2, respectively. Only *p* values < 0.05 are shown. All the other *p* values, statistical analyses and source data are provided as a Source Data file.

(Fig. 2B). In contrast, the neutralization breadth index (NBI), calculated as neutralization capacity against the alpha, beta and gamma-specific VOCs in relation to the neutralization of the WT virus[11,16], showed a significant increase over time (Fig. 2C). These data support the idea that in both children and adults, post-infection sera can recognize VOCs, and such recognition of VOCs can improve over time thanks to a progressive maturation of the immune response in recovered individuals.

**Progressive increase in plasmablasts and IgA-positive cells over 12 months after infection**

The distribution and frequency of B cell subpopulations in children and adults show fundamental differences[17] that can influence the individual immune response to infection. Extensive B cell phenotyping of SARS-CoV-2 seropositive adults and children 4 months and 12 months post infection (Fig. 3A–C, Supplementary Fig. 4A) showed higher percentages of B cells in children than in adults (Supplementary

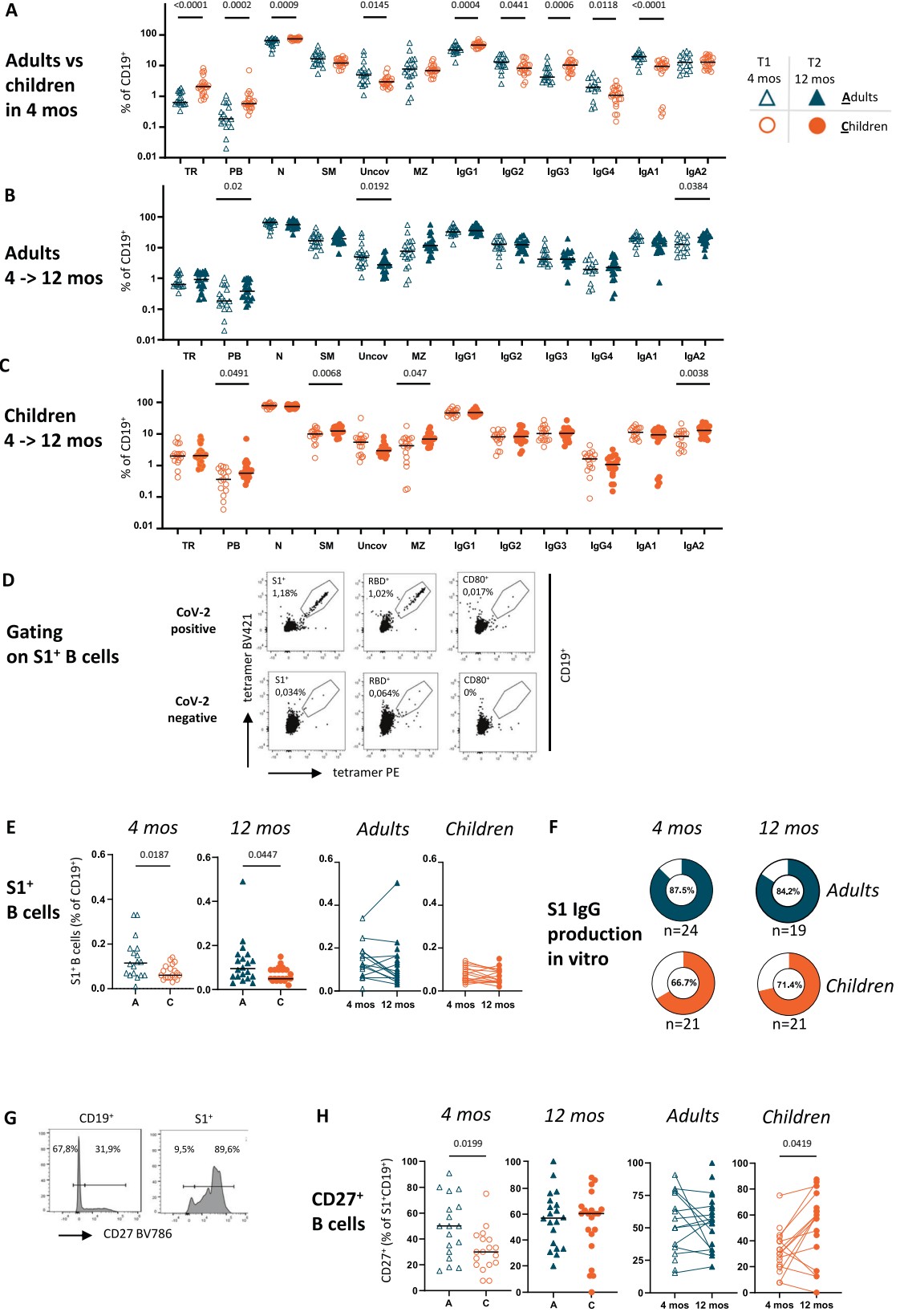

Fig. 4B). Within the B cells, children had higher transitional and naïve B cells but lower class switched memory cells (Fig. 3A), reflecting a less experienced immune system. Within the class-switched B cells, IgG1 and IgG3 were more frequent and IgG2, IgG4, IgA1, and IgA2 were less frequent in children than in adults (Fig. 3A). Over the 8-month observation period, most of the B cell subpopulations remained stable

(Figs. 3B, C), with the exception of an increased proportion of plasmablasts and IgA2-positive and unconventional (IgD⁻CD27⁻) B cells in adults (Fig. 3B). In children, we observed an increase in plasmablasts, switched memory cells, marginal zone-like B cells and IgA2-positive B cells (Fig. 3C). These data may support the concept of ongoing B cell activation in infected individuals, with the appearance of more mature

**Fig. 3 | S1-specific B cells are low but stable in frequency and progressively acquire a memory phenotype in children.** Only participants who were seropositive at T1 (for definition of seropositivity, see Methods) are plotted.
**A−C** Detailed B-cell phenotypes were determined at T1 (4 months after diagnosis; empty symbols) and T2 (12 months after diagnosis; filled symbols) in adults (**A**; blue symbols) and children (**C**; orange symbols). The detailed gating strategy is shown in Supplementary Fig. 4. **D** Gating strategy for the detection of S1$^+$ B cells using specific tetramers depicted for SARS-CoV-2 seropositive and seronegative donors.
**E** S1$^+$ B cells for both age groups and time points. **F** Proportion of participants with

proven production of S1 IgG (measured with Euroimmun ELISA) after stimulation of mononuclear cells with TLR9 agonist in vitro at T1 and T2. **G** Histograms showing an example of the detection of S1$^+$CD27$^+$ B cells. **H** Proportion of CD27$^+$ cells out of S1$^+$ B cells measured with flow cytometry at T1 and T2. The Mann-Whitney test and Wilcoxon matched-pairs signed rank test were used for comparing median values (black lines) between adults and children and between T1 and T2, respectively. Only $p$ values < 0.05 are shown. All the other $p$ values, statistical analyses and source data are provided as a Source Data file.

B cells over time. As samples were acquired during a period of social distancing, we propose that the progressive differentiation of B cells is largely due to the original infection and not caused by re-infections in the examined individuals.

### Persistent low frequency of circulating S1-specific memory B cells in children

The frequency of circulating SARS-CoV-2-specific B cells was assessed in our cohort by S1 and RBD tetramer staining (Fig. 3D, Supplementary Fig. 5A, B). In contrast to the serum S1-specific antibody titer, S1-specific B cells were less frequent in children than in adults at both 4 and 12 months after infection (Fig. 3E). RBD-specific B cells were less frequent in children 4 months after infection, but the frequency in children and adults was not significantly different 12 months after infection (Supplementary Fig. 5B). S1-specific and RBD-specific B cells persisted at a similar frequency during our observation period (Fig. 3E, Supplementary Fig. 5B). While the frequency of tetramer-specific B cells within the B cell pool was stable in children and adults, we observed a progressive maturation of children's B cell response, as indicated by the acquisition of the marker CD27 in the S1-specific and RBD-specific B cells (Fig. 3G, H, Supplementary Fig. 5C). This indicates that in children, a longer time period was needed for S1-specific B cells to acquire a memory B cell phenotype, as indicated by CD27 expression. Stimulation via TLR9 favors blasting and antibody production from memory B cells, while naïve B cells are only poorly stimulated and do not undergo class switching[18]. We studied the ability of S1-specific and NCP-specific memory B cells to produce specific antibodies upon TLR9 stimulation (Fig. 3F, Supplementary Fig. 6). In vitro activation resulted in the secretion of S1-specific antibodies in more than 80% of adults with a history of SARS-CoV-2 infection (Fig. 3F). In line with the lower frequency of S1-specific B cells in children, fewer children showed S1 antibodies in the supernatant of stimulated B cell cultures (Fig. 3F). This assay was highly specific, as we did not find any S1 or NCP-specific antibody production in vitro in noninfected individuals (Supplementary Fig. 6A). The frequency of S1-specific memory B cells detected by flow cytometry correlated with immunoglobulin concentrations in the culture supernatant in adults (Supplementary Fig. 6B), in line with the model that detection of specific B cells by tetramer staining corresponded to the presence of memory cells that can be rapidly reactivated. In children, the frequency of S1-specific B memory cells correlated with antibody secretion only at 12 months after infection (Supplementary Fig. 6B). This may be due to the low frequency of S1-specific memory B cells 4 months after infection or to their immature phenotype. In the supernatant of stimulated B cell culture, we also tested the presence of S1 IgA antibodies (Supplementary Fig. 6C) and NCP IgG antibodies (Supplementary Fig. 6D). The number of adults carrying S1 IgA memory cells that could be reactivated decreased over the observation period, while S1 IgA memory cells increased in children (Supplementary Fig. 6C), similar to the increase in serum S1-specific IgA. In contrast, the NCP response remained stable (Supplementary Fig. 6D).

### Durable T cell response in adults and children 12 months after SARS-CoV-2 infection

After infection with SARS-CoV-2, virus-specific CD8$^+$ and CD4$^+$ T cells can be studied by tetramer staining in peripheral blood[19–22]. The

detection of specific T cells by tetramers is limited by the restriction of the HLA-type. As our cohort was not selected by a specific HLA type, we assessed the presence of SARS-CoV-2-specific CD4$^+$ and CD8$^+$ T cells by restimulation of T cells with a SARS-CoV-2 peptide mix in vitro, followed by analysis of the activation markers CD69 and CD137 (Fig. 4A, B, Supplementary Fig. 7A, F). This activation-induced quantification assay allowed the detection of SARS-CoV-2 S1-reactive T cells with very high sensitivity, as only one child out of 48 seropositive participants had T cell numbers below the cut-off threshold (Supplementary Fig. 7C, see methods for details). The assay was also highly specific, as S1-reactive T cells were detected in only one non-infected, seronegative participant out of 46 and in none of 25 donors whose material was preserved before 2019 (Supplementary Fig. 7C and ROC curve in Supplementary Fig. 7D, E). Specific CD4$^+$ and CD8$^+$ T cell responses were detectable in both children and adults 4 and 12 months after infection, albeit at a lower frequency in children (Fig. 4C, D). The frequency of specifically activated CD4$^+$ cells remained constant over the 8-month observation period in adults, while a significant reduction in specific CD4$^+$ T cell frequencies was observed in children (Fig. 4C). CD8$^+$ T cells were also detected at a lower frequency in children than in adults but persisted in both age groups over time (Fig. 4D). Similar results were found after stimulation with a peptide mix covering the C-terminal part of the S protein with homology with endemic coronaviruses (SARS-CoV-2 S2-peptide mix, Supplementary Fig. 8). In contrast, when T cells were stimulated with a S1- and S2-peptide mix of four different common coronaviruses ("pan-corona") (Supplementary Fig. 9A), T cell activation was similar between children and adults. Additionally, stimulation with an adenovirus-derived peptide mix resulted in equal proportions of virus-specific CD4$^+$ and CD8$^+$ T cells in adults and children (Supplementary Fig. 9B). In contrast, stimulation with CMV peptides resulted in significantly lower specific T cell activation in children, in line with the known lower CMV infection rates in children (Supplementary Fig. 9C). Thus, a stronger recall T cell response in convalescent adults than in children seems to be a specific feature of SARS-CoV-2 infection.

### Similar distribution of effector memory T cells in children and adults but an altered cytokine profile in children

As for the B cell compartment, the T cells in children were largely naïve and showed low frequencies of central and effector memory T cell subsets (Supplementary Fig. 10), in line with previous observations[23,24]. We analyzed the phenotype of SARS-CoV-2-specific T cells in both cohorts. Despite the expected differences in unstimulated bulk CD4$^+$ T cells with age-dependent distribution of naïve and memory subpopulations (Supplementary Fig. 10), within the population of SARS-CoV-2-specific CD4$^+$ T cells, we found a similar distribution of naïve, effector memory and terminal effector memory cells between children and adults, and only CD4$^+$ central memory T cells were slightly lower in frequency in children 4 months after exposure (Fig. 5A–C). SARS-CoV-2-specific CD8$^+$ T cells showed a similar distribution of naïve and memory subpopulations in adults and children (Fig. 5D–F). The large majority of S1-specific CD4$^+$ and CD8$^+$ T cells were effector memory T cells, as expected in a virus-specific immune response[24]. The phenotypes of specific CD4$^+$ and CD8$^+$ T cells remained constant 4 months and 12 months postinfection in adults (Fig. 5B, E). In children, we

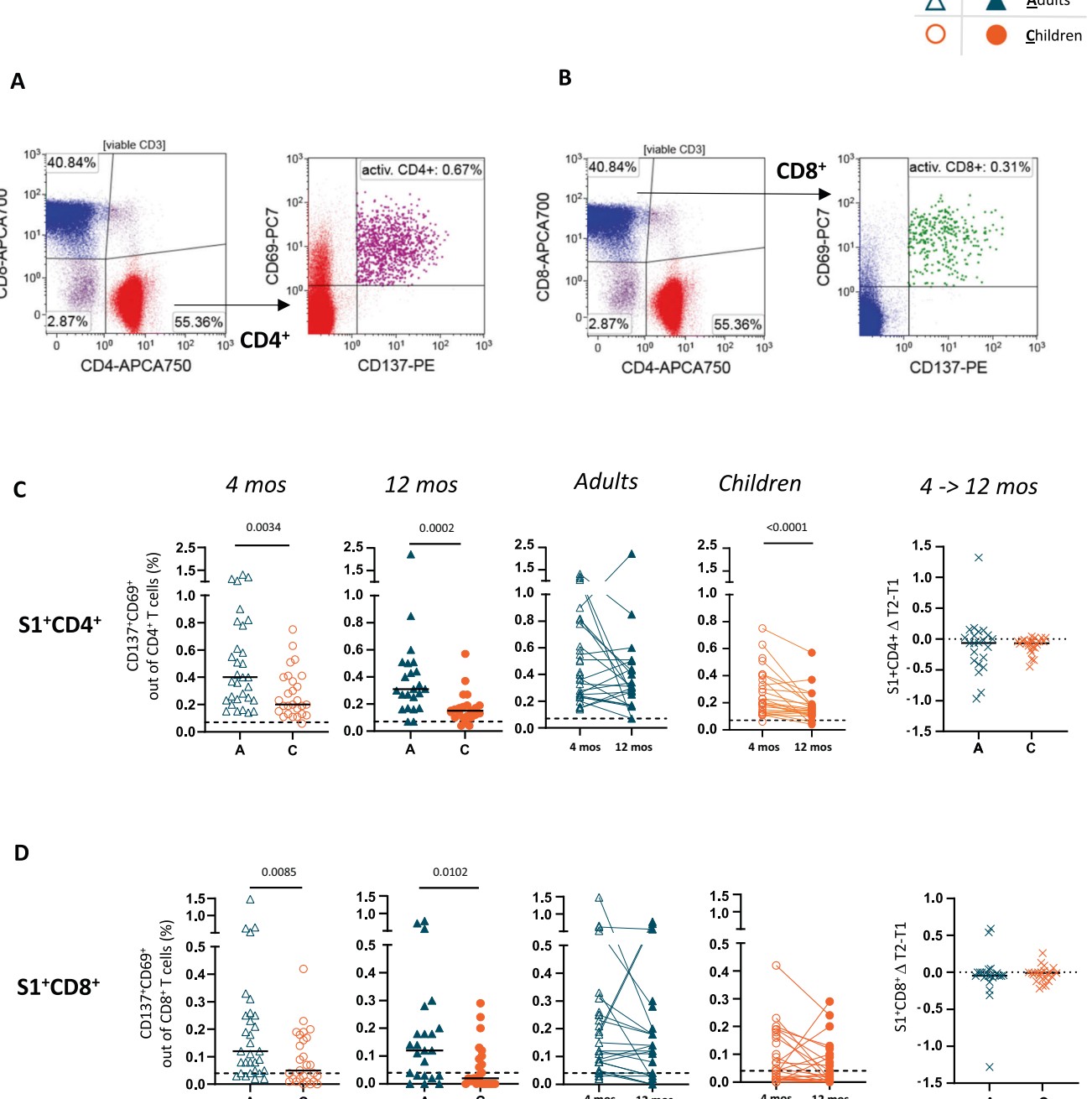

**Fig. 4 | Low but stable S1 Specific T cell response one year after SARS-CoV-2 infection in children.** The SARS-CoV-2-specific T cell response was analyzed using an activation-induced marker (AIM) assay by flow cytometric detection of CD69⁺CD137⁺ T cells after stimulation with SARS-CoV-2 Spike 1 peptide mix. Representative dot plots of CD69⁺CD137⁺CD4⁺ (**A**) and CD8⁺ (**B**) T cells are shown. The results for CD4⁺ (**C**) and CD8⁺ (**D**) of seropositive adults (blue) and children (orange) at 4 months (open symbols) and 12 months (filled symbols) after infection are shown. The dotted line in the slope graphs depicts a null difference. Black dashed lines indicate the limit of detection of 0.07% (CD4⁺) and 0.04% (CD8⁺). The Mann-Whitney test and Wilcoxon matched-pairs signed rank test were used for comparing median values (black lines) between adults and children and between the two timepoints, respectively. Only *p* values < 0.05 are shown. All the other *p* values, statistical analyses and source data are provided as a Source Data file.

observed a progressive decrease in effector memory CD4⁺ (Fig. 5C) and CD8⁺ (Fig. 5F) T cells and a progressive increase in CD8⁺ terminal effector T cells (Fig. 5F).

To evaluate the functionality of SARS-CoV-2-specific T cells, we analyzed cytokine secretion in the supernatant of restimulated T cells (with an S1-specific peptide mix) from patients 4 months postinfection. We assessed Th1 cytokines (IFN-γ and TNF), that are important for virus

clearance), proinflammatory cytokines (IL-1β, IL-17A) and cytokines regulating inflammatory reactions (IL-10, IL-13). IFN-γ is the primary cytokine in SARS-CoV-2-specific T cells. Comparison of the cytokine profiles in adults and children showed reduced IFN-γ, TNF, IL-10, IL-17A and IL-1β in T cells from children versus adults but similar levels of IL-13 (Fig. 5G). The reduced inflammatory cytokine secretion may contribute to the reduced symptoms observed in children compared to

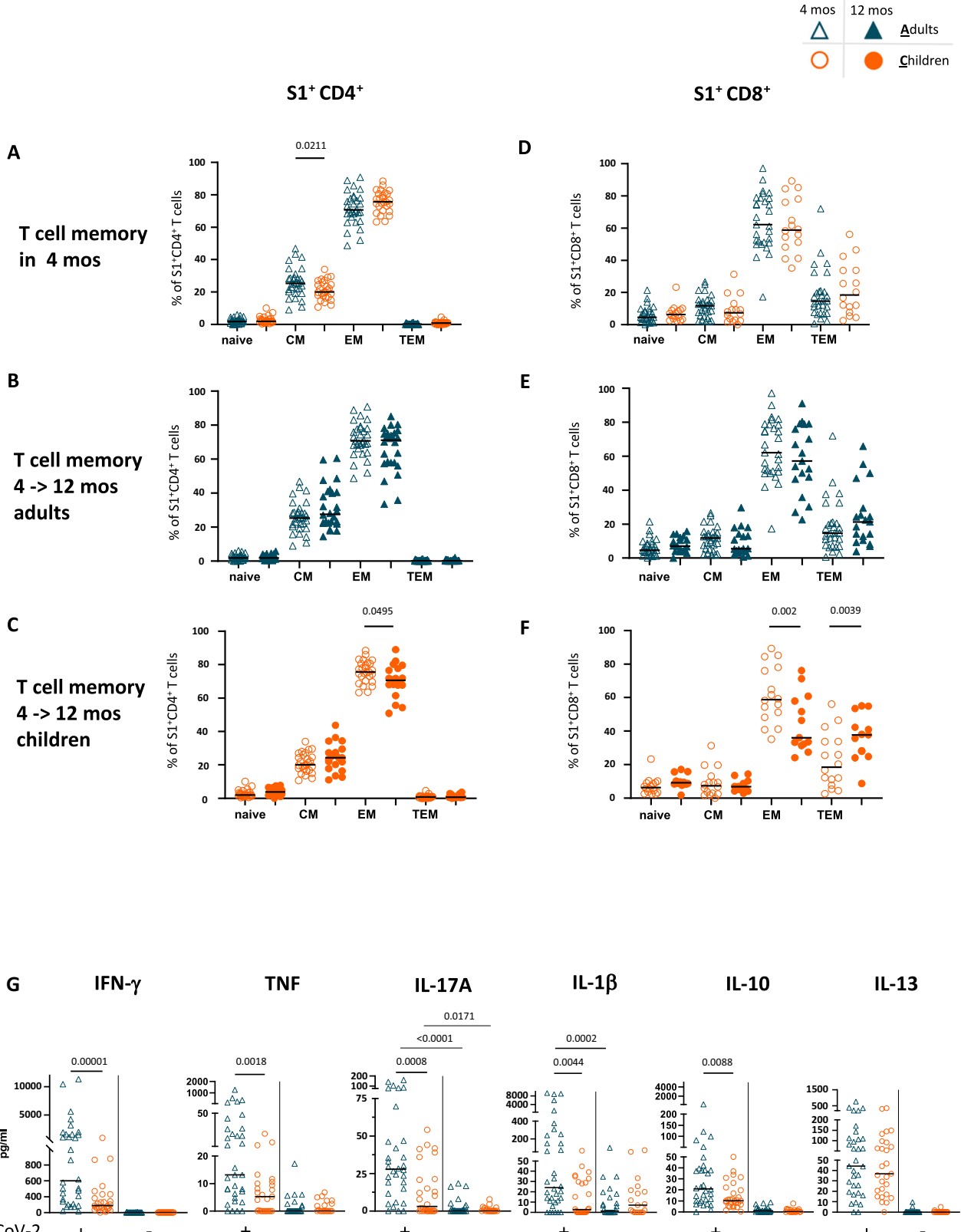

adults. While IFN-γ, TNF and IL-10 levels correlated with the percentage of S1-specific T cells among peripheral mononuclear cells (PBMC), this was not the case for the inflammatory cytokines IL-1β and IL-17A (Supplementary Fig. 11). Notably, children's T cells were able to secrete cytokines comparable to adults in response to pancoronavirus antigens (Supplementary Fig. 12).

**Persistence of cellular immune response and reduction of serological response up to 12 months post infection**

To gain a global view of the persistence of SARS-CoV-2-specific immunity and the progressive changes in the B and T cell compartments, we generated a heat map representing the log2 fold change in the ratio between the parameters measured 12 and 4 months after

**Fig. 5 | Phenotype and function of S1-specific T cells.** Distribution of T cell sub-populations: specifically stimulated CD4+ (**A**–**C**) and CD8+ (**D**–**G**) T cell sub-populations. CD69+CD137+ (=S1 specific) T cells of seropositive individuals were differentiated according to their expression of CD45RO and CCR7 in naïve T cells (CD45RO−CCR7+), central memory (CM) T cells (CD45RO+CCR7+), effector memory (EM) T cells (CD45RO+CCR7−) and terminal effector memory (TEM) T cells (CD45RA−CCR7−). **A, D** The T cell subpopulations among S1+ T cells of adults (blue symbols) and children (orange symbols) 4 months after SARS-CoV-2 infection are shown. The Mann-Whitney test was used to compare median values (black lines) between adults and children. CD4+ (**B, C**) and CD8+ (**F, G**) T cell subpopulations of S1+ T cells assessed at 4 months (open symbols) are shown in comparison to those at 12 months (filled symbols) in adults (blue) and children (orange). The Mann-Whitney test was used to compare median values (black lines) between the two timepoints. Statistical significance was defined as *$p \leq 0.05$, and only significant differences are indicated. **H** Detection of released cytokines in the supernatant of S1-stimulated mononuclear cells in vitro. The amount of IFN-γ, TNF, IL-10, IL-13, IL-17A and IL1β in the supernatant of stimulated cells of SARS-CoV-2 seropositive (CoV-2+) and SARS-CoV-2 noninfected (CoV-2−) family members was assessed by a multiplex immunoassay. The Mann-Whitney test was used to compare median values (black lines) between adults and children. Only *p* values < 0.05 are shown. All the other *p* values, statistical analyses and source data are provided as a Source Data file.

infection (Fig. 6A). This allowed for comparison of the changes in each parameter on a unified scale. This global analysis indicated that while specific humoral immunity (characterized by antibody response against S1, RBD and NCP) decreased over time, cellular immunity, such as S1-specific and RBD-specific B cells and S1-specific CD4+ T cells, was mostly stable. However, there were exceptions: S1-specific IgA responses in serum increased in children and some adults; S1-specific CD8+ T cells decreased in several individuals, primarily in children. Both the specific B and T cell compartments showed dynamic maturation over 12 months, especially in children, with acquisition of CD27 on specific B cells and acquisition of the terminal effector memory phenotype in specific CD4+ and CD8+ T cells in both children and adults. Hence, the reduction in humoral immunity paralleled the maturation of cellular immunity even in the absence of reinfection or vaccination. This concept is further supported by the progressive increase in peripheral plasmablasts, marginal zone-like cells and IgA2+ class-switched B cells. Additionally, the dynamics of the immune response in children and adults were mostly similar, with the exception of progressive IgA class switching and later expression of CD27 in specific S1 cells.

Looking for interdependence, we performed a matrix analysis using the Spearman test for significant correlations between the parameters 4 and 12 months after infection. Four months after infection (Fig. 6B), neutralizing antibodies positively correlated with the specific humoral response as well as the presence and maturation status of SARS-CoV-2-specific B cells in children. The presence of a specific antibody response and specific B cells positively correlated with total plasmablasts and switched memory cells and inversely correlated with naïve cells. These data suggested that the reduced cellular immunity might result from immaturity of the immune system in children. In adults, all specific antibody responses correlated with each other, as well as the specific B cell responses 4 months after infection. This suggested an independent development of plasmablasts and memory cells in the adult immune system. Interestingly, class switched CD27-negative unconventional memory B cells, which contain B cells that develop at extrafollicular regions[25] inversely correlated with specific antibody responses and the memory phenotype (CD27+) of specific B cells in adults. As described previously, the extrafollicular response is expanded in COVID-19[12], and extrafollicular cell fate favors plasmablast development with low affinity at the expense of germinal center-dependent maturation.

Twelve months after infection (Fig. 6C), correlation plots between children and adults converged, with a strong correlation between specific antibody and B cell responses. Notably, in children, S1-specific CD4+ T cells correlated with S1-specific and RBD-specific memory B cells and their maturation into CD27+ memory cells, in line with the known function of CD4 T cells in B cell development. At 12 months after infection, we were able to measure RBD-specific IgG in the saliva of the participants in our cohort. We found that in children, the presence of RBD IgG in the saliva positively correlated with serum RBD and S1 IgG as well as their neutralizing capacity, and it inversely correlated with unconventional memory cells in children. In the adult

cohort, RBD IgG in the saliva also correlated with specific serum antibody responses but also with specific B and CD4+ T cells. These data suggest that high serum immunoglobulin concentrations may correspond to protective levels of virus-specific IgG at mucosal surfaces, which represent entry sites for SARS-CoV-2, both in children and adults. In the adult population, we found a correlation between the specific serum antibody responses to S1 and RBD-specific memory B cells (Figs. 6B, C) both 4 and 12 months after infection, suggesting a parallel development of plasma cells and memory cells. In contrast, in children, the serum antibody response did not significantly correlate with the presence of S1-specific or RBD-specific memory cells at 4 and 12 months (Figs. 6B, C). As antibodies are secreted in the serum by plasmablasts and plasma cells, these data suggested a predominant plasmablast response in children.

## Discussion

We report the persistence of the specific SARS-CoV-2 humoral and cellular immune response in children and adults over 12 months following mild or asymptomatic COVID-19. Children showed a dominant antibody and plasmablast response and a lower frequency of specific memory B and CD4+ and CD8+ T cells with reduced secretion of proinflammatory cytokines 4 months after infection. In the following 8 months, the frequency of specific B and T cells remained stable in children, but the cells changed their phenotype, showing evolving maturation, in line with enhanced neutralization potency and breadth of specific antibodies in both adults and children.

The children's immune responses to SARS-CoV-2 at 4 months differed from those in their parents with higher serum antibody titers but with a lower frequency of S1 or RBD-specific memory B cells. While in adult sera, antibody titers correlated with the frequency of specific memory B cells, this was not the case in children. These data suggest that in children, antibody-secreting B cells and plasma cells dominate the SARS-CoV-2 immune response, while conventional memory cells constitute only a minimal part of the B cell response. The dynamics of the adaptive immune response upon infection or vaccination in children are understudied. Available data upon vaccination for rubella and measles show a predominant antibody response compared to non-secreting memory in children compared to adults[26], supporting the idea that plasmablast development and antibody secretion may be favored in children under specific circumstances. Indeed, children generally show a lower frequency of peripheral memory B cells[27]. Conventional memory cells require germinal center maturation[28–30] and CD4+ T cell help[31,32]. Hence, the reduced number of S1-specific memory B cells could be a consequence of the lower frequency of specific CD4+ T cells in children. In line with this hypothesis, the maturation of S1-specific and RBD-specific memory cells into CD27+ positive cells correlated with S1-specific CD4+ T cells in children in our cohort.

Memory B cells contribute to long-term viral immunity[33]; they can quickly respond upon re-encounter with the pathogen and contribute to specific antibody secretion. Memory B cells can also contribute to organ-specific immunity at pathogen entry sites, such as the lung or

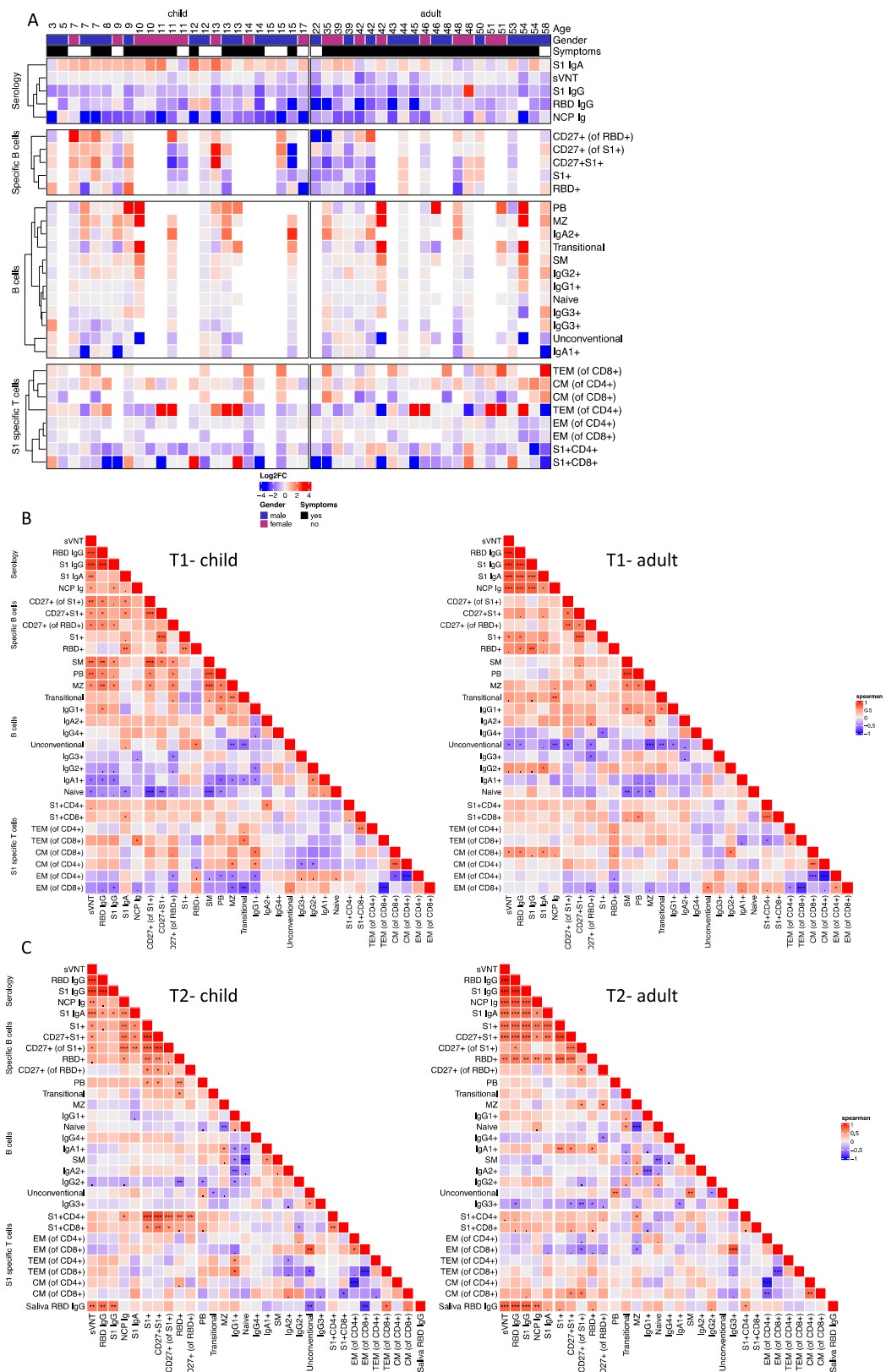

the gut[34]. We found functional memory B cells specific for SARS-COV-2 that were reactivated by polyclonal stimuli such as TLR9 activation in more than 80% of adults and in more than 60% of children with a history of infection. These data reinforce the predominant antibody response in children and confirm the function of memory B cells detected by tetramer staining.

The B cell response changed dynamically up to 12 months after infection. In the adults of our family cohort, we observed a significant reduction in S1-specific, RBD-specific and NCP-specific IgG antibodies over an 8-month observation period, accompanied by a significant increase in neutralization potency and by the ability to neutralize VOCs[12]. Indeed, a similar decrease in S1 and RBD antibody titers has

**Fig. 6 | Mapping of the longitudinal adaptive immune response in children and adults after SARS-CoV-2 infection.** Only participants who were seropositive at T1 (for definition of seropositivity see Methods) are plotted (22 adults, 24 children). **A** Heatmap depicting the log2-fold change in the ratio of T2 (12 months after SARS-CoV-2 infection) versus T1 (4 months) for the SARS-CoV-2-specific antibody response, B- and T cell subsets, and (S1 and/or RBD)-specific B and T cells. Red values indicate an increase, whereas blue values indicate a decrease. Missing values are depicted with white color. The annotations at the top show the age and sex of the probands and history of any COVID-19-related symptoms. The color scale was limited to values ranging from −3 to 3. Two pairs of heatmaps depict Spearman correlation between the selected parameters (s. above) for children (left) and adults (right) for T1 (**B**) and T2 (**C**). Significance is indicated within the heatmap ($p < 0.001$: ***$p < 0.01$: **$p < 0.05$: * the markers are clustered based on the children's correlation within each specified group. Hence, there is no dendrogram depicted for the adult heatmap. Source data are provided as a Source Data file.

been shown in several adult cohort studies both with mild and severe COVID-19[8,35–38] as well as an increase in the proportion of neutralizing antibodies[10,39] and their ability of serum antibodies to recognize VOCs[39]. Importantly, we observed a similar dynamic of the immune response in children. Even though the specific IgG antibodies for S1, RBD and NCP decreased over time, the neutralizing potency increased. Moreover, children developed specific S1 IgA over 8 months. We observed changes not only in the antibody response but also in the phenotype of memory B cells. S1 and RBD-specific B cells that were present at significantly lower frequencies in children changed their phenotype and acquired CD27 expression, a marker for memory B cells[25]. These data support the idea that the immune response to the virus persists over 12 months and improves over time in children. The changes in the immune response to SARS-CoV-2 after infection might be, for instance, due to the persistence of either virus particles or virus antigen in the gut[40]. The increase in IgA in children may support this finding, as mucosal class switching is the major source of serum IgA[41]. However, literature data on adults with follow-up up to 6–9 months show that while S1 IgG content in the saliva is stable over time and correlates with serum IgG content, this is not the case for S1 IgA, which rapidly disappears in the saliva[42,43]. Unfortunately, because of the limitation of sample availability, we were not able to measure S1-specific IgA content in saliva to prove whether the IgA saliva content in children differs from that in adults.

For specific T cell responses, a lower frequency of SARS-CoV-2-specific CD4[+] and CD8[+] T cell responses in children was reported during the acute phase of the infection[3,44,45] and convalescence[3]. Children also show reduced responses to polyclonal (phorbol-12-myristat-13-acetate/ionomycin) activation[3]. We did not observe differences between adults and children in the immune responses to pan corona peptide stimulation. Hence, in our data, the reduction in the T cell response in children seems to be specific for SARS-CoV-2. On the other hand, a stronger local innate immune response in children was reported[46]. Together with the stronger antibody response, this may support the quick resolution of the infection in children, which in turn could lead to lower T cell activation and T cell memory production. Our findings may reflect differences in the response to new antigens in a less experienced immune system in comparison to an adult immune system, which has been exposed to different pathogens during its lifetime. Lower in vitro T cell activation was also reported in T cells of healthy children after staphylococcal enterotoxin B (SEB) stimulation together with age-dependent expression of the cytokines IFN-γ, IL-17A, Granzyme B, TNF and IL-2[47]. Similar to severe COVID-19, fatal outcomes in SEB-induced toxic shock syndrome (TSS) are the result of a cytokine storm. In addition, in this case, children have a lower TSS-associated mortality than adults. Reduced systemic T cell activation and cytokine release, mainly of inflammatory cytokines, may contribute to the mild course of SARS-CoV-2 infection in children. Additionally, a coordinated humoral and cellular immune response is beneficial for COVID-19 patients[48], so the demonstrated good correlation of IgG S1 serum antibodies with the proportion of S1-specific CD4 T cells is in line with the overall mild course of SARS-CoV-2 infection in our family cohort.

There are several reports about stable SARS-CoV-2-specific T cells up to 8 months after infection in adults[8,19,36,49,50]. In line with two recently published reports on children[3,4], we detected a stable proportion of SARS-CoV-2-specific CD4[+] and CD8[+] T cells in both children and adults up to 12 months after infection. Only a mild decrease in CD4[+] T cells was found, which was statistically significant in children. Although SARS-CoV-2-specific CD8[+] T cells are found to be less frequent than CD4[+] T cells, as shown by other groups[48,51,52], they evolve to the more mature terminal effector memory phenotype in children during convalescence, while in adults, the differences are less pronounced and not significant.

The major study limitation originates from the recruitment approach, as the data on the PCR tests for SARS-CoV-2 as well as the information on the clinical symptoms have been obtained retrospectively. Another bias could be introduced by the selection of families with at least one SARS-CoV-2-infected adult or child. Furthermore, the extent of the analysis was limited by the amount of material that could be obtained from the pediatric participants. In contrast, the inclusion of children and adults from the same household, i.e. the inclusion of seronegative household members as well-matched controls constitutes a remarkable strength of this study. This approach limits the environmental and genetic heterogeneity that can influence the variability of the immune response. Additionally, the prospective longitudinal analysis of the complex immune response in children up to one year after SARS-CoV-2 infection is unique. Finally, as the infected individuals suffered exclusively from mild or asymptomatic infections, the data can be extrapolated to the majority of SARS-CoV-2 infections in the community.

In conclusion, we have demonstrated that children mount a robust SARS-CoV-2 antibody response and a low but detectable memory B and T cell response after mild or asymptomatic SARS-CoV-2 infection that is stable up to 12 months post-infection. This response is characterized by progressive S1-specific memory B and T cell maturation as well as tuning of the humoral immune response demonstrated by a continuous increase in neutralization potency and class switching favoring protection of mucosal surfaces. Whether these data reflect a general characteristic of the immune system in children or a specific response feature in SARS-CoV-2 infection remains unclear. Our findings are also relevant for vaccination strategies in children. They support the concept of booster vaccination after primary infection with SARS-CoV-2 to consolidate the cellular memory response.

## Methods
### Study details
This study examined 28 households with PCR-confirmed or seropositive symptomatic SARS-CoV-2 individuals. Only households with at least one adult or one child with a history of SARS-CoV-2 infection were included. We analyzed 50 children, of whom 30 were positive and 20 were negative for SARS-CoV-2. Of the 63 analyzed adults, 37 were positive and 26 were negative for SARS-CoV-2. Written informed consent was obtained from adult participants and from parents or legal guardians on behalf of the children at both blood sampling time points. Participants were asked to donate a blood sample and complete a questionnaire at both time points. Children's views on giving a blood sample were respected throughout. This study was initiated by the University Children's Hospitals in Freiburg, Heidelberg, Tübingen and Ulm and approved by the independent ethics committee of each

center. Blood samples and data for this substudy were collected at study site Ulm in July 2020 and March 2021. Epidemiological and serological data describing the larger cohort have been published previously[13,14,53].

The study is registered at the German Clinical Trials Register (DRKS), study ID 00021521, conducted according to the Declaration of Helsinki, and designed, analyzed and reported according to STROBE guidelines.

## Participants

Participants were recruited during the first wave of the pandemic (May to July 2020) by local health authorities, public announcements and an in-hospital database of families/households with at least one confirmed PCR-positive individual. Donors were eligible for enrolment if they met the following inclusion criteria: (i) children (male or female) aged 0 to 18 years; (ii) parents and other adults (male or female) living in the same household with the investigated children (without age limit); (iii) residency in the state of Baden-Württemberg, Germany; and (iv) written consent for the study. Key exclusion criteria were (i) severe congenital diseases (e.g., infantile cerebral palsy, severe congenital malformations); (ii) congenital or acquired immunodeficiency; and (iii) no comprehension of the German language.

Samples were collected at two separate time points, an early time point (T1) approximately 4 months post-symptom onset and a late time point (T2) approximately 12 months post-symptom onset (Table 1).

At T2, 23 families took part, of whom we analyzed samples of SARS-CoV-2-positive individuals (24 children and 22 adults) for a second time.

Children and adults within eligible households were asked to provide information on demography and the presence of symptoms (fever, cough, diarrhea or dysgeusia) in relation to SARS-CoV-2 infection within the household or around the time of a positive PCR. The four symptoms listed above were chosen based upon previous research and public health advice given at the time. A "symptomatic case" was defined as a seropositive individual showing at least one of these four symptoms. All other seropositive cases that did not show any of these four symptoms were defined as asymptomatic. To qualify as negative control, i.e. SARS-CoV-2 naïve, the participant had to fulfil all the following criteria: (a) absence of SARS-CoV-2 PCR positive swab in personal history; (b) absence of COVID-19 symptoms temporarily related to SARS-CoV-2 infection within the family; (c) seronegative status at T1 (4 months) after SARS-CoV-2 infection within the family.

## Serological assays

Antibodies against SARS-CoV-2 were detected using the following three assays: (1) EuroImmun-Anti-SARS-CoV-2 ELISA IgG (S1), (2) Siemens Healthineers SARS-CoV-2 IgG (RBD), and (3) Roche Elecsys Ig (Nucleocapsid Pan Ig). Seropositivity was defined as any two of the three SARS-CoV-2 assays being positive.

## EuroImmun Anti-SARS-CoV-2 ELISA

The EuroImmun Anti-SARS-CoV-2 ELISA (IgG, IgA) was performed according to the manufacturer's instructions to detect IgG antibodies against the S1 domain of the SARS-CoV-2 spike protein. All samples used in the final analysis were measured with this assay. All samples were processed with the specified controls and calibrators. Serological analysis was performed blinded to all clinical covariables.

## Siemens Healthineers SARS-CoV-2 IgG

The Siemens Healthineers SARS-CoV-2 IgG (sCOVG) assay was performed according to the manufacturer's instructions on an Advia Centaur XPT platform to detect IgG antibodies against the receptor-binding domain (RBD) of the SARS-CoV-2 spike protein. All samples used in the final analysis were measured with this assay. All samples were processed with the specified controls and calibrators. Serological analysis was performed blinded to all clinical covariables.

## Roche Elecsys electrochemiluminescence immunoassay

The Roche Elecsys electrochemiluminescence immunoassay (ECLIA) was performed according to the manufacturer's instructions on a Cobas e411 or e811 platform to detect IgG, IgA and IgM antibodies against the nucleocapsid of SARS-CoV-2. All samples used in the final analysis were measured with this assay. All samples were processed with the specified controls and calibrators. Serological analysis was performed blinded to all clinical covariables.

## Anti-SARS-CoV-2 neutralizing antibody titer serologic assay kit

Samples were analyzed for neutralization with the surrogate SARS-CoV-2 neutralization test (GenScript; sVNT) as per the manufacturer's instructions and as published previously[54]. Briefly, samples and controls were incubated with an HRP-conjugated RBD fragment. Following this, the mixture was added to wells of a capture plate coated with human ACE2 protein. The plate was then washed three times to remove any complexes or nonbound antibodies. TMB was added and then stopped with the addition of a stop reagent. The plate was then read by a microtiter plate reader at 450 nm. The absorbance of the sample is inversely correlated with the amount of SARS-CoV-2 neutralizing antibodies. Positive and negative controls served as internal assay quality controls. The test was considered valid only if the OD450 for each control fell within the respective range (OD450 negative control > 1.0, OD450 positive control < 0.3). For final interpretation, inhibition rates were calculated as follows: Inhibition score (%) = (1 − (OD value sample/OD value negative control) × 100%). Scores <30% were considered negative, and scores ≥30% were considered positive.

## Anti-SARS-CoV-2 neutralizing antibody titer serologic assay kit

The principle of this competitive anti-SARS-CoV-2 neutralizing antibody titer serologic assay kit (ACRO Biosystems; sVNT*) is ELISA reaction that mimics the virus neutralization process and qualitatively detects anti-SARS-CoV-2 antibodies, which suppress the interaction between receptor binding domain (RBD) fragments of the viral spike (S) protein and the angiotensin-converting enzyme 2 (ACE2) protein bound to the surface of a microtiter plate. Samples and controls were diluted 1:10 according to the protocol and were added, together with horseradish peroxidase (HRP)-conjugated RBD fragment (HRP-RBD), to a capture plate coated with human ACE2 protein. Any unbound HRP-RBD or HRP-RBD bound to nonneutralizing antibodies was captured on the plate. Then, four washing steps were performed to remove complexes of neutralizing antibodies and HRP-RBD that did not bind to the plate. Subsequently, TMB was added as a substrate, allowing HRP to catalyze a color reaction. The color change from blue to yellow after the addition of the stop reagent was read on a microtiter plate reader at 450 nm (OD450). The absorbance of the sample is inversely correlated with the amount of SARS-CoV-2 neutralizing antibodies. Positive and negative controls served as internal assay quality controls. The test was considered valid only if the OD450 for each control fell within the respective range (OD450 negative control > 0.8, OD450 positive control < 0.1). For final interpretation, inhibition rates were calculated as follows: Inhibition score (%) = (1 − (OD value sample/OD value negative control) × 100%). Scores <30% were considered negative, and scores ≥30% were considered positive. To investigate the capacity of neutralizing antibodies to cover certain variants of interest (VOC), different modifications of HRP-RBD were used: SARS-CoV-2-spike RBD N501Y (B.1.1.7), SARS-CoV-2-spike RBD E484K, K417N, N501Y (B.1.351) and SARS-CoV-2-spike RBD E484K, K417T, N501Y (P.1). The assay protocol was followed as described above.

## Detection of anti-SARS-CoV-2 RBD antibodies in saliva

After saliva collection with Salivette (SARSTEDT, Germany), the sample was centrifuged at 1000 g for 2 min and stored within one hour after sampling at −20° until measurement. Nunc MaxiSorp 96-well microtiter plates (Thermo Fisher Scientific) were coated overnight at 4 °C with SARS-CoV-2 Spike S1 Protein RBD, AA 319-541, tag-free Catalog No.: P2020-022 (trenzyme GmbH, Konstanz, Germany), 2 µg/well in Coating Buffer pH 7,4 (CANDOR Bioscience GmbH, Wangen, Germany). Well contents were discarded by tapping and blocked with 200 µL of PlateBlock (CANDOR Bioscience GmbH) for two hours at room temperature. Afterwards, 4 washing steps with 300 µL of Washing Buffer TRIS (CANDOR Bioscience GmbH) were carried out by utilizing a microplate washer (ELx50 Microplate Washer, BioTek). The plates were blocked with the coating stabilizer Liquid Plate Sealer (CANDOR Bioscience GmbH) to stabilize the coated RBD, which contains structural epitopes. For this, 200 µl of Liquid Plate Sealer was pipetted into each well and incubated for 4 min at room temperature. Afterwards, the wells were emptied without washing. Saliva samples were diluted 1:4 in SafetyTector (CANDOR Bioscience GmbH), and 100 µl was pipetted into each well. In addition, a standard series with an antibody (anti-SARS-CoV-2 Spike S1(RBD)-AB, Clone CR3022 Antikörper online, Aachen, Germany) was applied to the plate. This antibody was previously calibrated to a WHO standard in a separate measurement series to enable quantification of the saliva samples. The antibody was used at the following concentrations: 0, 9.1, 17.8, 44.9, 79.8, 152.3, 299.0, and 613.3 mIU/ml. Samples and standards were incubated for 2 h at room temperature on a shaker (Titramax 1000, Heidolph Instruments GmbH & Co. KG, Kelheim, Germany). The plates were washed 4 more times with Washing Buffer TRIS. Then, 100 µl of HRP-labeled goat anti-human IgG (Jackson ImmunoResearch, distributed by Dianova, #109-035-098) diluted to 1 µg/ml with HRP-Protector (CANDOR Bioscience GmbH) was added to each well and incubated at room temperature for 1 h. Afterwards, the plates were washed again 4 times with Washing Buffer TRIS. Then, 100 µl SeramunBlau slow2-50 (Seramun Diagnostica GmbH, Heidesee, Germany) was added to each well for the reaction at room temperature for 5 min. The reaction was stopped by the addition of 50 µl of 2 N H2SO4. The optical density at 450 nm was measured using a microplate reader (Epoch SN1607257, BioTek). The calculation of the measured values was performed with Gen5 version 2.09 software (BioTek).

## Investigation of B cells

**Tetramerization of S1 and RBD.** B cell-specific tetramers were prepared as described[55]. A biotinylated form of recombinant S1 and RBD proteins (BioLegend) was tetramerized by the addition of PE-conjugated or BV421-conjugated streptavidin (BioLegend) and used for B cell tetramer staining assays. Briefly, SA-PE or SA-BV21 was added in an amount equal to 1/5 of the monomer substrate amount. SA was added in 5 equal portions to the monomer and incubated each time at 4 °C for 20 min on a shaker. Protease inhibitor was added to the tetramers at a 1x final concentration (Sigma). The tetramers were filled up to 100 µl with 0.1% BSA in PBS and stored at 4 °C.

**Flow cytometry.** The phenotypes of the donors' PBMCs were determined by flow cytometry (Cytek Aurora, Cytek) with the antibodies listed in the Supplementary Table (Table S1). Dead cell exclusion was performed using a Zombie NIR Fixable Viability Kit (Biolegend).

**In vitro PBMC activation and ELISA.** PBMCs were plated at $10^6$ cells/ml and incubated for 9 days left unstimulated or stimulated with CpG (TCGTCGTTTTGTCGTTTTGTCGTT) and hIL-2 (Immunotools). The supernatants of the in vitro culture and serum of the vaccinated donors were used to determine the presence of SARS Cov-2 IgG antibody (Euroimmun). Total immunoglobulin (Ig) isotype concentrations were quantified by ELISA. Briefly, 96-well plates (Nunc Maxisorp) were coated with anti-human IgG/A/M (Jackson ImmunoResearch) in bicarbonate buffer. Bound Ig was detected with alkaline phosphatase-conjugated anti-human IgM, IgG and IgA (Jackson ImmunoResearch) and developed with p-nitrophenyl phosphate (Sigma-Aldrich) in DEA buffer. Ig concentrations were calculated by the interpolation of calibration curves generated by using an Ig standard (N Protein Standard SL; Siemens). To determine SARS Cov-2-specific IgM, an S1 precoated plate (Euroimmun) was developed as described for total Ig.

## Investigation of T cells

**Detection of SARS-CoV-2-specific T cells by activation-induced marker assay.** Ten to thirty milliliters of peripheral blood was collected in heparin tubes, and PBMCs were isolated by density gradient separation (Pancoll separating solution, Pan-Biotech) and cryopreserved for batched analysis.

Activation-induced marker (AIM) assays were performed using a modified approach as described previously[52,56,57]. PBMCs were thawed in complete RPMI cell culture medium (PAN Biotech), 100 U/ml penicillin/streptomycin, 2 mM L-glutamine and 14 mM HEPES (all Gibco, Thermo-Fisher Scientific) containing 10% FCS (PAN Biotech). After washing twice (PBS, Gibco, Thermo Fisher Scientific), the cells were resuspended at a concentration of $1 \times 10^7$/ml in complete RPMI cell culture medium containing 10% human AB serum (PAN Biotech). Cells were seeded at $10^6$/100 µl/well in a 96-well U-bottom plate and stimulated for four days at 37 °C/6.5% CO2 with 1 µg/ml SARS-CoV-2 spike protein peptide-pools: 1. PepMix SARS-CoV-2 spike glycoprotein pool 1 (JPT Peptide Technologies), containing overlapping peptides of the N-terminal part (residues 1-643) of the SARS-CoV-2 structural spike glycoprotein, in this study named "Spike 1" and 2. PepMixTM SARS-CoV-2 spike glycoprotein pool 2 (JPT Peptide Technologies), containing overlapping peptides of the C-terminal part (residues 633-1273) of the SARS-CoV-2 structural spike glycoprotein, in this study named "Spike 2". Additionally, cells were stimulated by a mix of peptide pools covering the complete spike region of common coronaviruses. For this purpose, spike 1 and 2 pools of Spike PepMixTM HCoV-OC43, HCoV-229E, HCoV-NL63 and HCoV-HKU1 were dissolved and pooled together to obtain 25 µg peptide/ 50 µl DMSO, which was used at 1 µg/ml for stimulation. Equimolar amounts of DMSO were used as a negative control. As a positive control, we used CEFX Ultra SuperStim Pool (JPT Peptide Technologies), which contains a peptide mixture of 17 different pathogens, including human herpesviruses, measles virus, rubella virus, *Haemophilus influenza*, human adenovirus 5, and *Clostridium tetani*.

To prove the specificity of SARS-CoV-2 stimulation, we used PBMCs from previously unexposed individuals as negative controls by using adult buffy coat PBMCs ($n = 15$) and routinely collected residual PBMCs from pediatric stem cell transplantation patients ($n = 10$) with nonimmunological and nonmalignant diseases (all gave written informed consent for research uses of the samples). All samples of unexposed individuals were collected and cryopreserved before 2019.

After four days of stimulation, the cells were washed with PBS and stained with Zombie-green Fixable Live/Dead Stain (Biolegend) for 10 min and RT in the dark, followed by staining with anti-human CD137-PE, CD69-PE Cy7, CCR7 Pacific Blue (all Biolegend), CD45RO ECD, CD279 (PD-1) PE-Cy5.5, CD3 APC, CD8 APC-AlexaFluor 700, CD4 APC-AlexaFluor 750, and CD45 Krome Orange (all Beckman-Coulter) for 20 min and RT in the dark (for antibody concentration see Supplementary Table 1). Stained cells were washed in IF medium (PBS/5% BSA), and data were acquired on a Beckman-Coulter 10 color NAVIOS Flow cytometer. Data analysis was performed using KALUZA Analysis 2.1. (Beckman-Coulter). The background values of CD137+CD69+ T cells obtained after DMSO incubation (Supplementary Fig. 7F) were subtracted from the values of CD137+CD69+ T cells after stimulation with peptide mixes to calculate the proportion of specifically activated T cells. The limit of detection of specifically activated cells was

calculated by assessing the mean +/− standard deviation of unexposed controls ($n = 25$) for CD4$^+$ (=0.07%) and CD8$^+$ (=0.04%) T cells.

**Detection and quantitation of cytokines in T cell stimulation cultures.** For quantitation and detection of IFN-γ, IL-1β, IL-10, IL-13, IL-17A and TNF in the supernatant of stimulated cells of the same samples used for AIM, we used a customized ProcartaPlex Multiplex Immunoassay (Invitrogen, Thermo Fisher Scientific). After stimulation and centrifugation of the cells, 100 μl of supernatant was collected in a new 96-well plate and cryopreserved. After thawing the supernatant on ice, the ProcartaPlex assay was performed according to the manufacturer's instructions. Briefly, after washing the beads twice in a 96-well plate using a magnetic plate washer, 50 μl supernatant was mixed with magnetic beads coupled to antibodies against cytokines and incubated on a microtiter plate shaker (500 rpm) for 120 min at RT. After washing twice, detection antibody was added and incubated for an additional 30 min on a microtiter plate shaker. After washing twice, the concentration of cytokines was analyzed using a Bioplex 200 (Bio-Rad).

**Data analysis and figure generation.** All data analysis was performed using Graph Pad Prism 9.0. The type of statistical analysis performed and (where appropriate) the subset of the study population used are listed in the figure legends. Between-group differences in continuous endpoints were analyzed using Mann-Whitney U tests, while correlations were analyzed using Spearman rank test. All analyses were exploratory in nature, and p-values may not be interpreted as confirmatory.

Heatmaps were generated using R version 4.1.2 using the ComplexHeatmap package (v 2.10.0). The tidyverse packages were used for data handling. For the log2-fold change heatmap, the color scale was manually limited to −3 to 3. If both values at timepoints 1 and 2 were 0, the LFC was set to 0. Correlation analysis was performed using the rcorr package using a Spearman correlation for the children and adults separately. The order of the values was determined by clustering the correlation matrix for the children. The same order was then applied to the correlation matrix of the adults. The heatmap was again plotted using the ComplexHeatmap package[58,59].

### Reporting summary
Further information on research design is available in the Nature Portfolio Reporting Summary linked to this article.

## Data availability
A short version of the study protocol is available at the German Clinical Trials Register (DRKS, www.drks.de), study ID 00021521. The full study protocol is available from https://www.drks.de/drks_web/navigate.do?navigationId=trial.HTML&TRIAL_ID=DRKS00021521. Individual participant data, including data dictionaries will not be available, since we did not seek parental consent for data sharing. These data are available from the corresponding authors upon request. Source data for each figure is provided with this paper.

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

## Acknowledgements

We would like to thank Yvonne Müller, Sandra Steinmann and Vanessa Missel for excellent organizational and secretarial support and Andrea Hänsler, Gudrun Kirsch, Ulrike Tengler, Ulrike Formentini, Sevil Essig, Alexandra Niedermayer, Ingrid Knape, Helgard Knaus, Andrea Schuster, Linda Wolf, Boram Song and Sonja Landthaler for excellent assistance with sample collection and storage. We are grateful to Anne Rensing-Ehl for input regarding T cell analysis. Alina Seidel and Rüdiger Groß are part of the International Graduate School in Molecular Medicine Ulm. This work was financially supported by the Ministry of Science, Research and Arts Baden-Württemberg within the framework of the special funding line for COVID-19 research to the Freiburg, Tübingen, Ulm and Heidelberg centers, and the Federal Ministry of Health to the Freiburg study site (P.H. and R.E.), the Federal Ministry for Economic Affairs and Energy on the basis of a decision by the German Bundestag COMBI-COV-2 (J.M. and P.R.), and DFG VO 673/5-1 (R.E.V.). R.E. is supported by the Berta-Ottenstein Programme for Advanced Clinician Scientists, Faculty of Medicine, University of Freiburg. M.R. is supported by DFG (SFB1160 project B02). The funders had no role in the study design, data collection, data analysis or the decision to publish.

## Author contributions

E.M.J., D.F., H.R., R.E., M.S., K.M.D., B.T., A.R.F., P.H., M.R. and A.J. conceived the study. E.M.J., D.F., G.S., P.R., M.R. and A.J. designed the experiments. H.R., R.E., M.S., J.M., F.K., R.E.V., B.T., A.R.F., P.H., K.M.D., M.R. and A.J. procured funding. E.M.J., D.F., F.T., M.C., R.L., I.J., F.S., J.S., A.N.D., A.Z., M.H., P.R., C.B., C.L., A.S., R.G. and A.J. performed the experiments. E.M.J., D.F., M.C., M.Z., C.B., S.F.N.B. and A.J. collected samples or organized their collection. T.S., G.S., B.J., and H.S. supported the sample collection and provided key resources. E.M.J., D.F., F.T., M.C., R.L., I.J., T.S., A.N.D., A.Z., M.H., P.R., C.B., L.C., J.M., F.K., A.S., R.G., M.R. and A.J. curated the data. E.M.J., D.F., M.C., M.H., B.M., M.R. and A.J. performed the data analysis. E.M.J., M.C., I.J., M.H. and A.J. generated the figures. E.M.J., D.F., M.R. and A.J. wrote the first draft of the manuscript. All authors approved the final version of the manuscript. All authors confirm that they had full access to all the data in the study and accept responsibility to submit for publication. M.R. and A.J. supervised jointly the project and serve both as corresponding authors.

## Funding

## Competing interests

The authors declare no competing interests.

## Additional information

[1]Department of Pediatrics and Adolescent Medicine, Ulm University Medical Center, Ulm University, Ulm, Germany. [2]Department of Rheumatology and Clinical Immunology, Medical Center University of Freiburg, Faculty of Medicine, University of Freiburg, Freiburg, Germany. [3]Institute of Virology, Ulm University Medical Center, Ulm, Germany. [4]CANDOR Bioscience GmbH, Wangen im Allgäu, Germany. [5]Department of Transfusion Medicine, Ulm University, Ulm, Germany. [6]Institute for Clinical Transfusion Medicine and Immunogenetics, German Red Cross Blood Transfusion Service Baden-Württemberg – Hessen and University Hospital Ulm, Ulm, Germany. [7]Institute for Immunodeficiency, Center for Chronic Immunodeficiency (CCI), Medical Center - University of Freiburg, Faculty of Medicine, University of Freiburg, Freiburg, Germany. [8]Department of Statistics, University of Ulm, Ulm, Germany. [9]Institute of Molecular Virology, Ulm University Medical Center, Ulm, Germany. [10]University Children's Hospital Tuebingen, Tuebingen, Germany. [11]Center for Pediatrics and Adolescent Medicine, Medical Center, Faculty for Medicine, University of Freiburg, Freiburg, Germany. [12]Department of Pediatrics I, University Children's Hospital Heidelberg, Heidelberg, Germany. [13]CIBSS - Centre for Integrative Biological Signalling Studies, Albert-Ludwigs University, Freiburg, Germany. [14]Institute of Immunology, Center for Pathophysiology, Infectiology and Immunology, Vienna Medical University of Vienna, Vienna, Austria. [15]These authors contributed equally: Eva-Maria Jacobsen, Dorit Fabricius. ✉e-mail: marta.rizzi@uniklinik-freiburg.de; ales.janda@uniklinik-ulm.de

