## [Peer Review File · Nature Communications]

High antibody levels and reduced cellular response in children up to one year after SARS-CoV-2 infectionREVIEWER COMMENTS

Reviewer #1 (Remarks to the Author):

The study focus on a small family cohort with at least one child or adolescent affected by SARS-CoV-2 in a longitudinal study in the baseline, 4 and 12 months after infection. The study evaluates the serum antibody response to SARS-CoV-2, neutralizing capacity against the original Wuhan (WT) strain, B cell, and T cell response change between adults and children respectively in the longitudinal course. I have two main comments as follows,

1. It is unclear what is the advantage of recruiting the family cohort in the study design and compared to children's responses to parents' response to SARS-CoV-2 affection.
2. A big flaw in the study is the statistical analysis. Since the data is longitudinal, it is recommended to adopt the mixed model to analyze the association between age group (child/adult) and immune response by controlling all the confounders. P-values for all associations should be reported.

Reviewer #2 (Remarks to the Author):

Jacosen and colleagues investigated humoral and cellular immune responses to COVID infection in children vs. adults, up to one year post mild/asymptomatic infection. This is an important question that has implications for the current pandemic and for vaccine recommendations in the pediatric age range.

The demographics, timepoints of sample collection, and infection status of adults and pediatric study subjects is well matched. The data are clearly presented and consistent with conclusions drawn. However there are a couple of major concerns, listed below:

1. Figure 1B: the text describes the increased serum IgA responses in children particularly at the 12 month post infection period - is this also the case for mucosal/secretory IgA? It would important to address this point to demonstrate maturation of B cell response
2. T cell responses in adults and children, lines 197-220 and Fig 4C. These data demonstrate CD4+ and CD8+ AIM percentages as other previous publications. However, an unstimulated condition is not shown and it is unclear if such background signal is accounted for. Also, typically these data is shown as % AIM cells from the parent CD4 or CD8 T non-naive population, given as the authors indicate, most of the responsive cells are TEM or TCM phenotype. hence, would recommend changing the display
3. For the same T cell assay data above, positive control data is not shown, such a tetanus toxoid stim, or a CMV peptide pool, which would be important to demonstrate the robustness of the assay for other antigens
4. The authors state the children demonstrate a PB dominant immune response, but Fig 3 does not demonstrate increased frequency of PB in children compared to adults
5. Line 402 - in the conclusion, the authors indicate that the children mount a robust immune response after mild/asymptomatic infection; however, throughout the paper, in comparison to adults, humoral and cellular responses are shown as decreased in children compared to adults - this conclusion should capture this summary

Reviewer #3 (Remarks to the Author):

The authors have presented a very interesting comparison in the long term immune responses to mild COVID-19 in children vs adults. The longitudinal household approach here provides an excellent method of control, and is a major strength for the study! The distinct differences between these groups provide insight into possible reasons for the different clinical progression observed in children vs adults, and has important implications for how exposure to COVID-19 might influence response to

exposure to new variants.

Major items:

1. Line 128: "Neutralizing antibodies persisted in both cohorts over time, with a minimal decrease in the adult population (Fig. 1C)" -- I'm not sure if minimal is entirely appropriate.
2. Line 137: the VOCs in Figure 2A are compared independently, but comparisons between them are made in the text. Perhaps adding some direct comparisons in the figures with stats would fulfil this.
3. Line 179: "We studied the ability of S1- and NCP-specific memory B cells to produce specific antibodies upon TLR9 stimulation (Fig 3F, S5)". -- No stats? I appreciate that this is presented as yes/no nominal data, but is there a more continuous quantitative measure to back it up?
4. Line 216-217: "data not shown" -- I think for this claim, some data would have to be included.

Minor items:

1. In the abstract 'maturation of the S1 specific...' is used, which refers to reduced titre and increased neutralising activity, but I feel this should be explicitly linked here (as in my first read, I thought these referred to two different observations).
2. Line 87-88: 'uninfected family members' I take were PCR-negative individuals? Perhaps this should be clarified here.
3. Line 90-91: this sentence should probably clarify that the observations occurred in both children and adults.
4. line 100-102: the wording here states that all infected patients were mild, and that some infected were asymptomatic. These should be mutually exclusive, and some clarity in the wording would help.
5. Line 134: specify IgG (as opposed to over immunoglobulin)
6. Line 157: "h the exception of an increased proportion of plasmablasts and IgA2 positive B cells" -- also unconv. B cells in Fig 3B.
7. Line 166-167: "by S1 or RBD tetramer staining (Fig. 3D, S4A-B)" should probably be 'and'.
8. Line 170: "...was similar 12 months..." should probably be stated as something along the lines of "was not significantly different"
9. Lines 243 and 246: "INF" I assume should be "IFN"

- Thomas Ashhurst

REPLY TO REVIEWERS' COMMENTS

Reviewer #1 (Remarks to the Author):

The study focus on a small family cohort with at least one child or adolescent affected by SARS-CoV-2 in a longitudinal study in the baseline, 4 and 12 months after infection. The study evaluates the serum antibody response to SARS-CoV-2, neutralizing capacity against the original Wuhan (WT) strain, B cell, and T cell response change between adults and children respectively in the longitudinal course. I have two main comments as follows,

1. It is unclear what is the advantage of recruiting the family cohort in the study design and compared to children's responses to parents' response to SARS-CoV-2 infection.

We think that the longitudinal investigation of household members after natural infection constitute a unique opportunity to reduce the genetic and environmental heterogeneity present in non-household studies. The other two reviewers support our opinion; one of them (#3) even highlights the design of the study as one of the major strengths of the study. We edited the weaknesses and the strengths of the study explaining the advantage of using related subjects.

2. A big flaw in the study is the statistical analysis. Since the data is longitudinal, it is recommended to adopt the mixed model to analyze the association between age group (child/adult) and immune response by controlling all the confounders. P-values for all associations should be reported.

We would like to thank the reviewer for this comment. Indeed, our study design follows a longitudinal approach with only two time points of measurement though. Of course, the analysis of such a pre-post design needs to account for the repeated measurements structure of the data, i.e., the dependencies of measurements clustered within each subject. This could either be done by using the proposed mixed model, or by means of the applied statistical hypothesis tests in a paired fashion. The latter, of course, presumes comparability of the pre- and post-measurement samples, whereas the mixed model would be additionally able to adjust for confounding, as suggested, and could deal with missing data. We believe that our approach of applying paired hypothesis tests is sufficient for this pre-post comparison of immune response parameters, since there is hardly any confounding factor to be necessarily accounted for during the analysis (see Table 1). The only minor imbalances which could be found refer to the distribution and frequency of infection symptoms within the cohort. However, as we pointed out in our manuscript, there is no association between the frequency of symptoms and the immune response. Further, there are only a few subjects for whom no post measurement is available, i.e., the number of missing data is negligible. Thus, we believe that there is no distinct advantage of the mixed model approach here, since there is hardly a potential confounding variable to be accounted for. The small sample size does not allow to apply cluster analysis of the positive individuals within the families.

We agree with the reviewer that the p-values should be reported transparently. For the sake of better readability of text, we have decided to place asterisk symbols into the figures and the corresponding p-values into the legends.

Reviewer #2 (Remarks to the Author):

Jacobsen and colleagues investigated humoral and cellular immune responses to COVID infection in children vs. adults, up to one year post mild/asymptomatic infection. This is an important question that has implications for the current pandemic and for vaccine recommendations in the pediatric age range. The demographics, timepoints of sample collection, and infection status of adults and pediatric study subjects is well matched. The data are clearly presented and consistent with conclusions drawn. However, there are a couple of major concerns, listed below:

1. Figure 1B: the text describes the increased serum IgA responses in children particularly at the 12-month post infection period - is this also the case for mucosal/secretory IgA? It would be important to address this point to demonstrate maturation of B cell response.

This is a quite relevant remark. Initially, we measured only IgG and, unfortunately, we do not have any saliva left to perform additional measurement of IgA. The previously published data show that the IgA secreted in saliva 9 months after SARS-CoV-2 infection are less persistent than IgG antibodies (Isho et al., Alkharaan et al.).

We added the following text in the manuscript (page 13):

“However, literature data on adults with follow-up up to 6-9 months show that while S1 IgG content in the saliva is stable over time and correlates with serum IgG content, this is not the case for S1 IgA, that rapidly disappear in the saliva 42,43. Unfortunately, because of limitation of sample availability, we were not able to measure S1 specific IgA content in saliva to prove whether the IgA saliva content in children differs from adults.”

References:

Isho, B. et al. Persistence of serum and saliva antibody responses to SARS-CoV-2 spike antigens in COVID-19 patients. *Sci. Immunol.* 5, (2020).

Alkharaan, H. et al. Persisting salivary IgG against SARS-CoV-2 at 9 months after mild COVID-19: A complementary approach to population surveys. *J. Infect. Dis.* 224, 407–414 (2021).

2. T cell responses in adults and children, lines 197-220 and Fig 4C. These data demonstrate CD4+ and CD8+ AIM percentages as other previous publications. However, an unstimulated condition is not shown and it is unclear if such background signal is accounted for.

We certainly agree on importance of comprehensive reporting of negative controls. We added the following sentence to the method section:

“Background values of CD137+CD69+ T cells obtained after DMSO incubation (see Fig S7F) were subtracted from values of CD137+CD69+ T cells after stimulation with peptide-mixes to calculate the proportion of specifically activated T cells.”

Additionally, we added the following sentence to figure legend of S6:

“DMSO control values, (shown in Fig S7F) were subtracted for the calculation of specifically activated T-cells.”

Also, typically these data is shown as % AIM cells from the parent CD4 or CD8 T non-naive population, given as the authors indicate, most of the responsive cells are TEM or TCM phenotype. Hence, I would recommend changing the display.

We have based the discussed displays on the seminal reports of Dan et al. and Cohen et al. Stimulated by the reviewer’s comment we searched the published literature and found that - similarly to our manuscript – the percentage of AIM cells is usually shown as all CD4+ or CD8+ T cells (Dan, J. M. et al *Science* 2021; .

Cohen, C. A. et al. Nat. Commun. 2021). We have not found any publication reporting on % AIM from the non-naïve population. One additional argument is that the children do have more naïve T cells and a still restricted repertoire of memory T cells, gating on the non-naïve population would probably show, that the proportion of SARS-CoV2 specific T cells within the memory T cells is higher in children as there are fewer memory T cells for other pathogens compared to adults.

To make this point clear, we adapted the sentence on page 9 as follows:

“Despite the expected differences in unstimulated bulk CD4+ T cells with age-dependent distribution of naïve and memory subpopulations (Fig. S10), within the population of SARS-CoV-2-specific CD4+ T cells, we found a similar distribution of naïve, effector memory and terminal effector memory cells between in children and adults, and only CD4+ central memory T cells were slightly lower in frequency in children 4 months after exposure (Fig 5A-C).”

References (in extenso) where % AIM positive cells are shown from the whole CD4 / CD8 compartment and the distribution of naïve/ memory T-cells within the AIM positive cells are cited in the manuscript.

References:

Dan, J. M. et al. Immunological memory to SARS-CoV-2 assessed for up to 8 months after infection. Science (80-.). 371, eabf4063 (2021).

Cohen, C. A. et al. SARS-CoV-2 specific T cell responses are lower in children and increase with age and time after infection. Nat. Commun. 12, 4678 (2021).

3. For the same T cell assay data above, positive control data is not shown, such a tetanus toxoid stim, or a CMV peptide pool, which would be important to demonstrate the robustness of the assay for other antigens.

We added plots with the results on T cell stimulation with adenovirus and CMV to the supplementary figure S9.

We added the following text on page 8 of the manuscript:

“In contrast, when T cells were stimulated with a S1- and S2-peptide mix of four different common coronaviruses (“pan-corona”) (Fig. S9A), T cell activation was similar between children and adults. Also stimulation with an adenovirus derived peptide-mix resulted in equal proportions of virus-specific CD4+ and CD8+ T cells in adults and children (Fig. S9B). On the contrary, stimulation with CMV peptides resulted in significantly lower specific T cell activation in children in line with the known lower CMV infection rates in children (Fig. S9C). Thus, a stronger recall T cell response in convalescent adults compared to children seems to be a specific feature of SARS-CoV-2 infection.”

4. The authors state the children demonstrate a PB dominant immune response, but Fig 3 does not demonstrate increased frequency of PB in children compared to adults.

Antibodies are secreted by plasmablast and plasma cells, while memory B cells do not secrete antibodies unless they are stimulated. In this work we are looking at the specific immune response to SARS-CoV2 that is not necessarily reflected by changes in the whole B cell compartment. In addition, long lived plasma cells home to the bone marrow few weeks after antigen encounter, and may not be found in peripheral blood. Indeed, we find high titers of SARS-CoV2 specific serum antibodies, indicating the presence of plasma cells secreting them, and low frequency of S1- and RBD-specific memory B cells. Based on this we draw the conclusion that the B cell response to SARS-CoV2 in children is mainly composed by plasmablasts and plasma cells secreting antibodies that are found in the serum. Looking at the global changes of the B cell compartment, in both adults and children we observed an increase in the frequency of peripheral

plasmablasts over the observation period, and in the children an increase in switched memory cell frequency. The second is not mirrored by an increase in the frequency of S1-specific cells, but in their phenotypic maturation (expression of CD27). Indeed, the expression of CD27 is part of the definition of the memory phenotype within the human B cell compartment.

We have adapted the sentence on page 11 as follows:

'As antibodies are secreted in the serum by plasmablasts and plasma cells, these data suggested a predominant plasmablast response in children.'

5. Line 402 - in the conclusion, the authors indicate that the children mount a robust immune response after mild/asymptomatic infection; however, throughout the paper, in comparison to adults, humoral and cellular responses are shown as decreased in children compared to adults - this conclusion should capture this summary

We have adapted the text in conclusion as follows:

"In conclusion, we have demonstrated that children mount a robust SARS-CoV-2 antibody response and a low but detectable memory B cell response after mild or asymptomatic SARS-CoV-2 infection that is stable up to 12 months post-infection."

Reviewer #3 (Remarks to the Author):

The authors have presented a very interesting comparison in the long-term immune responses to mild COVID-19 in children vs adults. The longitudinal household approach here provides an excellent method of control, and is a major strength for the study! The distinct differences between these groups provide insight into possible reasons for the different clinical progression observed in children vs adults, and has important implications for how exposure to COVID-19 might influence response to exposure to new variants.

Major items:

1. Line 128: "Neutralizing antibodies persisted in both cohorts over time. In the adult, with a minimal decrease in the adult population (Fig. 1C)" -- I'm not sure if minimal is entirely appropriate.

We have adapted the wording on page 6 as follows:

"Neutralizing antibodies were still detectable 12 months post infections in both cohorts, even though they significantly decreased in the adult population (Fig. 1C)."

2. Line 137: the VOCs in Figure 2A are compared independently, but comparisons between them are made in the text. Perhaps adding some direct comparisons in the figures with stats would fulfil this.

We have added supplemental Figure 3 showing these data.

3. Line 179: "We studied the ability of S1- and NCP-specific memory B cells to produce specific antibodies upon TLR9 stimulation (Fig 3F, S5)". -- No stats? I appreciate that this is presented as yes/no nominal data, but is there a more continuous quantitative measure to back it up?

In supplemental figure 6B we show how the amount of S1 IgG specific antibodies secreted by the in vitro restimulated memory cells correlate with the frequency of S1 memory cells in the adults both at 4 and 12 months observation time and in children only at 12 months. This is a quantitative representation of specific IgG secretion in the supernatant. In the adapted Supplemental Figure 6A a comparison of amounts of secreted S1 IgG, S1 IgA and NCP Ig in restimulated PBMCs supernatants between SARS-CoV-2 infected and non-infected individuals is shown. We could demonstrate high specificity of this assay as there are no SARS-CoV-2 specific antibodies detected in non-infected individuals. In supplemental figure 6A the concentration of the antibodies in the supernatant is shown on the Y axes and ranges from 0-10 RU. As we

stimulated total peripheral blood mononuclear cells in this protocol (PBMCs), the measure of the S1 specific IgG is proportional to the frequency of the 'functional' switched memory cells. Therefore, we plotted the frequency of patients with detectable antibodies upon restimulation as a qualitative (yes/no) measure (now supplemental figure 6C and D), as this indicates the real frequency of patients with 'functional memory'. The children at 4 months not only have a low frequency of specific memory cells, but also these cells are not very efficient in producing antibodies once they are restimulated, and this situation improved 12 months after infection, when the correlation between frequency and antibody secretion become significant.

All this is already explained in the main text.

4. Line 216-217: "data not shown" -- I think for this claim, some data would have to be included.

We have added Supplemental Figure 8 and adapted the text accordingly:

"Similar results were found after stimulation with a peptide mix covering the C-terminal part of the S protein with homology with endemic coronaviruses (SARS-CoV-2 S2-peptide mix, Fig. S8)."

Minor items:

1. In the abstract 'maturation of the S1 specific...' is used, which refers to reduced titer and increased neutralizing activity, but I feel this should be explicitly linked here (as in my first read, I thought these referred to two different observations).

The maturation of the immune response is referred to several observations: the enhanced neutralizing activity and breadth of the SARS-CoV2 specific antibodies, the progressive appearance of S1 IgA antibodies, the expression of CD27 on S1 specific memory cells and the more mature phenotype in specific T cells. We changed the text as follows:

"However, in children one year after infection increase in S1-specific IgA class switch and expression of CD27 on S1-specific B cells and T cell maturation were observed. These together with enhanced neutralizing potential and breath of the specific antibodies suggested a progressive maturation of the S1-specific immune response."

2. Line 87-88: 'uninfected family members' I take were PCR-negative individuals? Perhaps this should be clarified here.

The definition of non-infected individuals (negative controls) is described in the Methods. The participant had to fulfill the following criteria: a) absence of SARS-CoV-2 PCR positive swab in personal history; b) absence of COVID-19 symptoms temporary related to the SARS-CoV-2 infection within the family; c) seronegative status at T1 (4 months) after SARS-CoV-2 infection within the family. We guess that this definition is too complex to be discussed in the introduction and we point to the method section.

3. Line 90-91: this sentence should probably clarify that the observations occurred in both children and adults.

Text was modified as follows:

"SARS-CoV-2-specific immunity persists for over 12 months both in children and adults and shows signs of phenotypic maturation in both the B and T cell compartments."

4. Line 100-102: the wording here states that all infected patients were mild, and that some infected were asymptomatic. These should be mutually exclusive, and some clarity in the wording would help.

The text has been adapted as follows:

'In 93.5% of infected adults and in 66.7 of infected children, the disease course was mild, and in 6.5% of adults and 33.3% of children, infection was classified as asymptomatic.'

5. Line 134: specify IgG (as opposed to over immunoglobulin)

The sentence has been modified as follows:

"Importantly, the differences in S1- and RBD-specific antibody levels between adults and children were not due to a change in total serum immunoglobulin levels (Fig. S2A, S2B)."

6. Line 157: "... the exception of an increased proportion of plasmablasts and IgA2 positive B cells" -- also unconv. B cells in Fig 3B.

Text has been adapted as follows:

"...exception of an increased proportion of plasmablasts and IgA2-positive and unconventional (IgD-CD27-) B cells in adults (Fig. 3B)."

7. Line 166-167: "by S1 or RBD tetramer staining (Fig. 3D, S4A-B)" should probably be 'and'.

We have adapted the sentence accordingly:

"The frequency of circulating SARS-CoV-2-specific B cells was assessed in our cohort by S1 and RBD tetramer staining (Fig. 3D, S4A-B)."

8. Line 170: "...was similar 12 months..." should probably be stated as something along the lines of "was not significantly different"

We have adapted the sentence accordingly:

"RBD-specific B cells were less frequent in children 4 months after infection, but the frequency was not significantly different 12 months after infection (Fig. S5B)."

9. Lines 243 and 246: "INF" I assume should be "IFN"

We apologize for the typo, indeed IFN was meant. We have corrected it accordingly.

REVIEWER COMMENTS

Reviewer #1 (Remarks to the Author):

The authors addressed my comments.

Reviewer #2 (Remarks to the Author):

The authors have satisfactorily addressed my concerns and comments, as well as the other reviewers. The edits and changes have improved the manuscript overall and I recommend it for acceptance.

Reviewer #3 (Remarks to the Author):

The authors have addressed my concerns, and I thank them for their efforts addressing them. No further concerns or changes required for publication.